# Estimation of district-level fertility using age-structured census data and assessment of spatial–socioeconomic differentials in Botswana, 2022

Tiro Theodore Monamo👤*, Kannan Navaneetham

Department of Population Studies, University of Botswana, Private Bag, Gaborone, Botswana

* mtirotheodore@gmail.com

## Abstract

### Background

Reliable subnational fertility estimates are critical for evidence-based demographic policy and planning. While Botswana's national fertility decline is well documented, less is known about district-level fertility variation and the demographic and socio-economic mechanisms underlying these patterns. This study aims to: (i) produce district-level fertility estimates using standard direct demographic procedures applied to census data; (ii) assess the consistency of direct estimates with established indirect and model-based fertility estimation techniques; and (iii) examine how women's age structure, child mortality, socioeconomic status, and marital patterns shape spatial fertility differentials across districts.

### Methods

The analysis uses data from the 2022 Botswana Population and Housing Census for women aged 15–49. Direct fertility estimation was conducted through the calculation of age-specific fertility rates (ASFRs) and total fertility rates (TFRs). These estimates were systematically compared with indirect and model-based approaches, including Rele (1967), Hauer et al. (2013), Ponnapalli and Soren (2018), and Hauer and Schmertmann (2020), to assess robustness and internal consistency. Data quality was evaluated using parity distributions and reports of recent births, with attention to potential biases arising from recall error, proxy reporting, and reference-period misclassification. Descriptive demographic analysis was used to assess the influence of age structure, child mortality, women's socioeconomic characteristics, and marital composition on district-level fertility variation.

**Data availability statement:** The data supporting the findings of this study are not publicly available due to ethical and confidentiality restrictions related to individual-level survey data. However, the data are available from Statistics Botswana upon reasonable request. Requests for access can be directed to Statistics Botswana at info@statsbots.org.bw. Additionally, the data may be provided by the authors upon reasonable request, subject to approval from Statistics Botswana.

**Funding:** The author(s) received no specific funding for this work.

**Competing interests:** The authors have declared that no competing interests exist.

## Results

Direct and model-based fertility estimates were closely aligned, with national TFR values ranging between 2.6 and 3.0 children per woman, confirming Botswana's advanced stage of fertility transition and the reliability of census-based estimation. However, substantial subnational heterogeneity was observed. Urban districts (e.g., Gaborone, Francistown, Jwaneng) exhibited below-replacement fertility, associated with higher female educational attainment, delayed entry into childbearing, and greater labour market participation. In contrast, predominantly rural districts (e.g., Ngamiland West, Barolong, Central Mahalapye) maintained higher fertility levels (approximately 3.5–4.5), linked to elevated child mortality, lower levels of women's empowerment, and persistent traditional reproductive norms. Districts undergoing socioeconomic transition displayed intermediate fertility profiles. Differences in age structure and child survival emerged as key demographic mechanisms reinforcing spatial fertility inequalities.

## Conclusion

By integrating standard direct demographic techniques with indirect and model-based validation, this study demonstrates the robustness of census data for subnational fertility estimation in Botswana. Persistent spatial and socioeconomic disparities highlight the need for differentiated policy responses: rural districts require intensified investments in child health, female education, and reproductive health services to sustain fertility decline, while urban areas must plan for the social and economic implications of sustained below-replacement fertility. Addressing these divergent trajectories is essential for achieving balanced and equitable demographic development in Botswana.

## 1. Introduction

Fertility remains a central driver of demographic change, shaping population structure, influencing the demand for social services, and determining long-term economic trajectories [1]. Reliable measurement of fertility at subnational levels is therefore critical for policy formulation in health, education, labour planning, and social development. Census age-structured data provide a valuable basis for direct estimation of age-specific fertility rates (ASFRs) and the total fertility rate (TFR). However, in many contexts, such direct estimates are rarely validated against alternative model-based or indirect approaches that incorporate demographic adjustments, smoothing techniques, and spatial priors. This limitation has implications for accuracy and comparability, particularly in countries where fertility levels and patterns display marked geographic variation.

Botswana provides an instructive case study in this regard. The country maintains a relatively advanced census system capable of generating district-level data on age and birth distributions [2]. These data reveal substantial within-country heterogeneity,

ranging from low fertility levels in urban centres to persistently higher fertility in rural districts [3]. Nevertheless, the direct estimation of fertility from census data is not without challenges. Incomplete birth reporting, age misstatement, in-migration of women of reproductive age, double-counting of births, and cultural or regional reporting biases may distort fertility estimates [4]. In small districts, where recorded births are often minimal, statistical instability further complicates the computation of reliable fertility measures. Moreover, assumptions of uniform mortality and recall-period biases can exaggerate fertility estimates, underscoring the need for careful methodological triangulation.

The demographic context in Botswana further underscores the importance of subnational fertility analysis. Fertility has undergone a pronounced transition over the past five decades. Research by Bainame and Letamo [5], using the Arriaga method to estimate fertility trends between 1971 and 2011, documented a sustained decline from 6.6 children per woman in 1981 to 3.3 in 2001, and further to 2.8 by 2011. These changes reflect broader socioeconomic transformations, including urbanisation, rising educational attainment, and increased participation of women in the labour force, factors consistently associated with lower fertility, particularly in urban areas [6,7]. However, national-level analyses obscure district-level variation that is essential for understanding heterogeneous fertility dynamics.

Socioeconomic determinants play a central role in shaping fertility behaviour. These include women's educational attainment, employment status, marital and cohabitation patterns, household living conditions, and access to health services, factors commonly recognised in demographic and public health frameworks, including classifications adopted by the World Health Organization (WHO). In Botswana, these determinants vary considerably across districts, reflecting unequal development trajectories and differential access to services, yet their interaction with fertility outcomes at the district level remains underexplored.

The 2022 Botswana Population and Housing Census (PHC) provides a timely opportunity to address these gaps. The census introduced major methodological innovations by being the first in Botswana to fully adopt digital technologies across cartographic mapping, household listing, and enumeration processes [8,9]. These innovations enhanced data completeness, internal consistency, and timeliness, strengthening the potential of the 2022 PHC for subnational demographic analysis while still necessitating careful evaluation of data quality.

Against this backdrop, the objectives of this paper are threefold. First, it seeks to generate transparent, district-level TFR estimates from the 2022 Botswana census using standard demographic procedures. Second, it compares these direct estimates with model-based fertility measures derived from indirect approaches, including Rele's method and the frameworks advanced by Hauer et al. (2013), Ponnapalli and Soren (2018), and Hauer and Schmertmann (2020). These methods typically rely on census-based indicators such as child–woman ratios (CWRs) to infer fertility levels in contexts where complete vital registration data are lacking. Third, the study examines spatial and socioeconomic fertility differentials across districts, focusing on associations with women's age structure, under-five mortality, educational attainment, employment, and marital patterns.

The contribution of this study is both methodological and substantive. Methodologically, it demonstrates the value of rigorous direct fertility estimation while embedding results within a transparent comparative framework that integrates established model-based approaches. Substantively, it provides empirical evidence on the drivers of fertility variation across Botswana's districts, highlighting the influence of socioeconomic and spatial determinants. The policy relevance is immediate: accurate and reliable district-level fertility estimates are indispensable for guiding reproductive health services, educational programming, and family planning interventions tailored to local needs [10].

## 2. Data and methods

### 2.1. Data source

This study draws on data from the 2022 Botswana Population and Housing Census (PHC), the sixth national census conducted by Statistics Botswana. The census provides the most comprehensive source of demographic and socioeconomic information for the country and serves as the primary data source for district-level fertility estimation in this paper.

Enumeration was conducted on a de facto basis, whereby individuals were recorded at their place of residence during the census period. The dataset includes detailed information on age, sex, births in the 12 months preceding the census, educational attainment, employment status, marital status, and household characteristics.

The analytic focus is on women aged 15–49 and children under five years of age, which form the basis for computing ASFRs, TFRs, and indirect fertility indicators. Socioeconomic variables were extracted at the district level to support the analysis of spatial fertility differentials.

## 2.2. Data quality assessment and mitigation measures

Given known limitations of census-based fertility estimation, several steps were taken to assess and mitigate potential data quality issues. First, internal consistency checks were conducted by examining parity distributions and reported births in the 12 months preceding the census to identify age heaping, underreporting, or irregular fertility patterns. Second, fertility estimates were aggregated across five-year age groups to reduce random fluctuation in districts with small numbers of births. Third, indirect and model-based fertility estimation techniques were applied alongside direct methods to triangulate results and assess robustness. Where discrepancies emerged, greater analytical weight was placed on estimates that demonstrated consistency across methods. Finally, district-level analyses were interpreted with caution, recognising the influence of migration and reporting biases that cannot be fully corrected within cross-sectional census data.

## 2.3. Methodological framework for estimating district-level fertility

This study employed both direct and indirect approaches to estimate fertility levels at the district level in Botswana. The direct estimation was undertaken by computing Age-Specific Fertility Rates (ASFRs) for women aged 15–49, grouped into conventional five-year intervals. The Total Fertility Rate (TFR) was then derived by summing the ASFRs across all age groups, representing the average number of children a woman would bear if exposed to current age-specific fertility patterns throughout her reproductive life. While this method is transparent and widely used in demographic research, it is also sensitive to census data quality, particularly completeness of birth reporting, accuracy of women's age distribution, and district-level population mobility. For this reason, the study triangulates these direct estimates with several established indirect techniques.

Among the indirect approaches, Rele's method [11] was applied as one of the foundational techniques in fertility estimation. This approach relies on the Child–Woman Ratio (CWR), calculated as the number of children aged 0–4 per 1,000 women of reproductive age (15–49). The method exploits the statistical relationship between the Gross Reproduction Rate (GRR) and the CWR in stable populations, adjusted for mortality using coefficients (α and β) corresponding to different levels of life expectancy. The TFR is then approximated as 2.05 times the GRR. Rele's method is particularly valuable in data-limited settings due to its minimal requirements and capacity to produce fertility estimates over two periods, using children aged 0–4 to represent births in the 2.5 years preceding the census, and children aged 5–9 for births in the preceding 7.5 years. Despite assumptions of population stability and uniform mortality, this approach has been widely validated in diverse contexts and provides a useful comparative benchmark for the Botswana data.

A second set of indirect estimates was derived using the Child–Woman Ratio approach advanced by Hauer et al. [12]. This method builds on the established empirical relationship between the General Fertility Rate (GFR) and the TFR, simplifying estimation by treating the reproductive span as a single interval. The implied TFR (iTFR) is calculated using the number of children aged 0–4 relative to women of childbearing age, thereby reducing data demands. The method's strength lies in its feasibility in low-data environments and its adaptability to census structures where detailed fertility histories are unavailable. Similarly, the technique proposed by Ponnapalli and Soren [13] was implemented, extending the use of CWRs to estimate not only TFR but also related measures such as Crude Birth Rate (CBR), General Fertility Rate (GFR), and marital fertility indicators. This approach incorporates child populations aged 0–9, total female populations, and marital status data, making it especially suitable for subnational analyses where vital registration is incomplete.

Its validation in the Indian census context underscores its robustness, and its minimal data requirements align well with Botswana's district-level demographic records [14].

Finally, the study applied the more recent methodological framework developed by Hauer and Schmertmann [14], which refines indirect fertility estimation by incorporating demographic information embedded in population pyramids. Their Extended Total Fertility Rate (xTFR) model adjusts traditional child–woman ratio approaches by accounting for the proportion of women concentrated in high-fertility age groups (25–34) relative to the broader reproductive span, thereby improving accuracy in populations with non-uniform age structures. Moreover, the adjusted xTFR+ variant integrates under-five mortality (q5) through a survival multiplier, correcting biases that arise in settings where child mortality remains significant. Given the observed fluctuations in Botswana's under-five mortality over the past decade, the xTFR+ provides an important correction to conventional indirect estimates. By employing this suite of direct and indirect approaches, the study establishes a comprehensive and comparative framework for estimating district-level fertility, allowing for robust validation of census-based measures and deeper insights into spatial–socioeconomic fertility differentials.

The integration of both direct and model-based estimation techniques is essential for ensuring that fertility measures used in planning are not only methodologically sound but also contextually reliable. District-level variations in Botswana reflect complex interactions between data quality, demographic behaviours, and socioeconomic conditions. By triangulating direct estimates with multiple indirect approaches, the study minimizes the risk of systematic bias and provides policymakers with more dependable fertility indicators. This methodological rigour enhances the utility of the findings for designing targeted reproductive health services, education programmes, and labour market policies that are sensitive to the distinct demographic realities of each district.

## 2.4. Measurement of the validity of district-level fertility estimates

To assess the validity of the direct district-level TFR estimates, a comparative validation exercise was undertaken against five established model-based fertility measures. For each district, pairwise comparisons were made between the direct census estimates and the corresponding model-based estimates. The strength of association was quantified using Pearson correlation coefficients, enabling an assessment of the degree of linear correspondence between the direct and indirect measures. Correlation values were interpreted in line with conventional thresholds, with coefficients ranging from 0.70 to 0.89 considered strong and those above 0.90 interpreted as very strong alignment [15]. In addition to correlation analysis, descriptive comparisons were performed to identify systematic differences across districts, with particular attention to deviations in urban versus rural settings. This combined approach allowed for both statistical validation of the direct estimates and substantive interpretation of areas where divergences may reflect underlying demographic or socioeconomic processes rather than methodological error.

## 2.5. Assessing the effects of mortality, women's socioeconomic status, and marital patterns on fertility

To examine the determinants of district-level fertility variation in Botswana, the study linked direct Total Fertility Rate (TFR) estimates from the 2022 census with district-level indicators of child mortality, women's socioeconomic status, and marital patterns. Mortality was measured using infant and under-five mortality rates (U5MR) obtained from census data, providing a proxy for survival risks that may influence fertility through replacement behaviour. Women's socioeconomic status was captured through two key variables: the proportion of women aged 15–49 with tertiary education and the proportion of women employed, both derived from census microdata. Marital patterns were operationalized as the percentage of women aged 15–49 who were married or cohabiting in each district. Particular attention was given to urban–rural contrasts, transitional districts, and outliers, in order to understand how interactions between mortality, education, employment, and marital behaviour shape fertility outcomes. This integrated approach allowed the study to situate district-level fertility patterns within a broader demographic transition framework while highlighting context-specific dynamics unique to Botswana.

## 2.6. Ethics considerations

This paper is based on the analysis of secondary data. Ethical considerations were addressed during the original data collection phases of the census conducted by Statistics Botswana. As per the Botswana Statistics Act of 2009, participation in national surveys and censuses is mandatory, and individuals were informed of this requirement. Additionally, the Act stipulates that refusing to answer any questions constitutes an offence. Consequently, no separate ethical consent process was undertaken for this study, as the necessary ethical protocols were adhered to during the initial data collection.

## 3. Findings

This section presents the study's findings in five interrelated components. First, it assesses the quality of fertility data by examining parity distributions and reports of recent births, providing an empirical basis for evaluating the reliability of census-derived fertility indicators. Second, it presents a comparative analysis of district-level fertility estimates generated using direct census-based methods and a range of indirect demographic models, highlighting patterns of convergence and divergence across estimation approaches. Third, the consistency of fertility estimates across methods is evaluated to assess their internal validity and robustness at the district level. Fourth, the analysis examines the influence of women's age structure on fertility outcomes, focusing on how variation in the distribution of reproductive-age cohorts shapes observed spatial fertility patterns. Finally, the section investigates the combined effects of child mortality, women's socioeconomic characteristics, and marital dynamics in explaining regional fertility differentials across Botswana's districts.

### 3.1. Evaluation of fertility data quality

**3.1.1. Assessment of parity data.** In population censuses, the first question on fertility typically concerns women's lifetime fertility, measured by the number of children ever born [4]. As with most self-reported demographic data, fertility information is subject to several potential reporting errors. One important issue is recall bias, which tends to increase with the respondent's age. Older women are more likely to omit births, particularly those that occurred early in their reproductive lives or children who subsequently died [16]. A second concern arises when fertility data are provided by proxy respondents rather than the mother herself. In such cases, the proxy may have incomplete knowledge of the woman's reproductive history, especially regarding children who reside elsewhere or those who have died. Both of these factors may lead to the under-enumeration of parities.

Despite these potential challenges, the 2022 Botswana Population and Housing Census parity data display an overall pattern that aligns with demographic expectations. Average parity increases steadily with women's age, reflecting a positive association between age group and the number of children ever born. This age gradient is consistent with biological and demographic realities, as older women, having had longer exposure to childbearing, report higher parities than younger cohorts. The observed trend supports the general reliability of the parity data as a basis for estimating district-level fertility in Botswana (Fig 1).

**3.1.2. Assessment of recent births data.** In addition to parity, the quality of data on recent births is critical for estimating fertility levels and patterns. Several potential sources of error may affect the accuracy of this information. First, there is uncertainty regarding the exact timing of births in relation to the census reference period. This uncertainty may lead to the misplacement of births, either by incorrectly including births that occurred before the reference period or excluding those that occurred within it. Such reference-period misclassifications can bias fertility estimates.

Second, migration and mortality dynamics may contribute to under-reporting. For instance, women who had a recent birth but subsequently died or migrated out of the household prior to enumeration will not be captured in the census data. Similarly, households that experienced dissolution after a birth occurred may also fail to report these events. These omissions result in an underestimation of recent fertility levels.

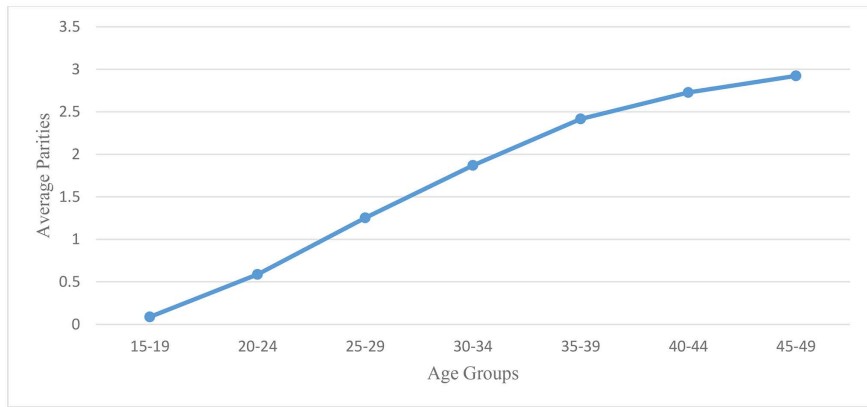

*Source: Authors' own calculations using results from Botswana's 2022 Population and Housing Census*

**Fig 1. Distribution of women by age and parity in Botswana, 2022.**

Despite these challenges, the reported age-specific fertility pattern in Botswana's 2022 Census data follows the expected demographic trajectory. Fertility rises steadily from ages 15–29, reflecting the concentration of childbearing in early adulthood. Fertility then declines progressively from ages 30–49, consistent with biological limits on fecundity and sociocultural shifts that typically accompany later reproductive ages. This pattern mirrors global evidence: for example, estimates from the American Society for Reproductive Medicine indicate that women under age 30 have approximately a 25% probability of conceiving naturally per cycle, declining to about 20% after age 30, and further reducing to around 5% by age 40 [17].

The alignment of the Botswana census results with established demographic expectations provides reassurance regarding the overall reliability of recent births data, despite the known limitations associated with recall, reporting, and household dynamics (Fig 2).

### 3.2. Comparative analysis of district-level fertility estimates

The district-level fertility estimates presented in **Table 1** highlight both consistencies and divergences across direct and indirect estimation techniques, underscoring the complexity of measuring fertility at subnational scales in Botswana. Across all districts, the direct estimates derived from the 2022 census generally align closely with model-based outputs, with national-level totals ranging from 2.6 to 3.0 across indirect methods and 2.9 from the direct approach.

Urban districts consistently record the lowest fertility levels, regardless of the method applied. For instance, Gaborone City's direct estimate of 1.8 children per woman closely matches indirect estimates (1.5–1.7), placing the capital well below replacement level fertility. Similar patterns are evident in Francistown (2.2), Jwaneng (1.8), and Sowa Town (1.4), where direct measures corroborate the persistent fertility suppression in urban and mining towns. Notably, however, small differences emerge across methods; for example, Sowa Town's direct estimate (1.4) is substantially lower than model-based outputs (2.0–2.9), likely reflecting data volatility in very small populations.

By contrast, rural districts show persistently high fertility, often exceeding 3.5 children per woman. Districts such as Ngamiland West (direct estimate 4.5), Central Tutume (3.7), Barolong (3.9), and Kweneng West (3.7) exemplify this pattern. Here, indirect estimates are generally consistent but occasionally understate fertility, suggesting that model-based approaches may not fully capture the reproductive intensity of rural populations. For example, Ngamiland West's direct estimate (4.5) is higher than any indirect method (3.8–4.4).

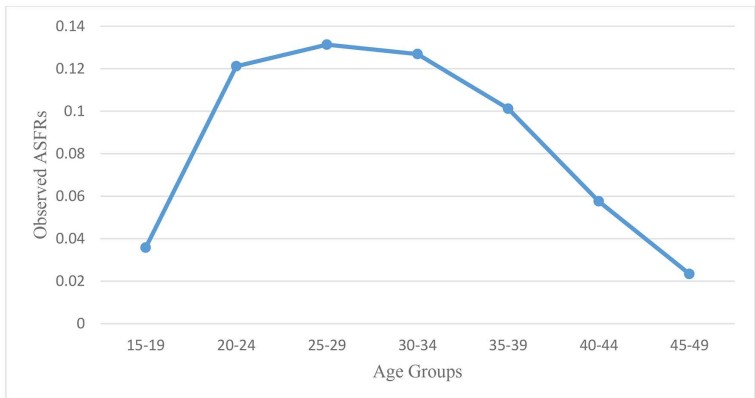

Source: Authors' own calculations using results from Botswana's 2022 Population and Housing Census

**Fig 2. Distribution of women by age and observed ASFRs in Botswana, 2022.**

Intermediate fertility levels are observed in semi-urban and peri-urban districts such as Kgatleng (2.9), Kweneng East (2.9), and Serowe/Palapye (3.1). Across these districts, the agreement between direct and indirect estimates is generally strong, with variations rarely exceeding 0.4 children per woman. At the national level, the convergence between direct (2.9) and model-based estimates (2.6–3.0) confirms that Botswana remains in the midst of its fertility transition, with district-level heterogeneity reflecting uneven progress across regions.

### 3.3. Validation of district-level fertility estimates

Validation of the district-level fertility estimates was undertaken by comparing the direct estimates derived from the 2022 Botswana Population and Housing Census with a set of established model-based estimates, namely the Rele TFR-1, Rele TFR-2, Hauer et al. (2013), Ponnapalli and Soren (2018), and Hauer and Schmertmann (2020). The results reveal a generally strong alignment across measures, with correlation coefficients ranging from 0.78 to 0.91. The strongest agreement was observed between the direct census estimates and Rele TFR-1 (2019) (r ≈ 0.90) and Hauer et al. (2013) (r ≈ 0.91), indicating that both recent indirect census adjustments and longer-term model-based projections captured the prevailing fertility patterns with considerable accuracy. Strong correlations were also observed with Ponnapalli and Soren (2018) (r ≈ 0.87) and Hauer and Schmertmann (2020) (r ≈ 0.88), further validating the internal consistency of the 2022 direct estimates. The weakest association but still strong, emerged with Rele TFR-2 (2014) (r ≈ 0.78) (See Fig 3).

### 3.4. Age structure of women of reproductive age and its influence on district-level fertility in Botswana

The percentage distribution of women across reproductive age groups provides a strong demographic basis for understanding district-level fertility variation in Botswana (See Table 2). Districts with high fertility levels consistently exhibit youthful reproductive age structures, where a large share of women are concentrated in the early and peak childbearing years. In high-fertility districts such as Ngamiland West (TFR 4.5), Barolong (3.9), and Central Tutume (3.7), women aged 15–24 constitute over 30% of the reproductive-age population, complemented by substantial proportions in the 25–29 age group.

In contrast, the lowest-fertility districts display an age distribution heavily weighted toward older reproductive cohorts. Urban districts such as Gaborone (TFR 1.8), Jwaneng (1.8), and Sowa Town (1.4) have a markedly different demographic profile, with less than 15% of women in the 15–19 age group and over 40% in the 30–44 range. Districts with moderate

**Table 1. Comparative district-level fertility estimates in Botswana, 2014–2022: Direct census-based and indirect model-based methods.**

| District | Settlement Type | Rele TFR-1 (2019) | Rele TFR-2 (2014) | Hauer et al. (2013) 2022 | Ponnapalli & Soren (2018) 2022 | Hauer & Schmert-mann (2020) 2022 | Direct Esti-mate (2022) |
|---|---|---|---|---|---|---|---|
| Gaborone City | Urban | 1.5 | 1.7 | 1.6 | 1.5 | 1.5 | 1.8 |
| Francistown City | Urban | 2.3 | 2.4 | 2.3 | 2.1 | 2.1 | 2.2 |
| Lobatse | Urban | 2.2 | 2.6 | 2.1 | 2.1 | 2.1 | 2.5 |
| Selebi-Phikwe | Urban | 2.3 | 3.1 | 2.2 | 2.4 | 2.3 | 2.3 |
| Orapa | Urban | 1.9 | 3.0 | 1.9 | 2.2 | 2.0 | 2.6 |
| Jwaneng | Urban | 1.8 | 2.4 | 1.8 | 1.9 | 1.7 | 1.8 |
| Sowa Town | Urban | 2.1 | 2.9 | 2.1 | 2.3 | 2.0 | 1.4 |
| South-East | Peri-Urban | 1.8 | 1.9 | 1.9 | 1.7 | 1.7 | 2.2 |
| Kweneng East | Peri-Urban | 2.5 | 2.5 | 2.5 | 2.2 | 2.3 | 2.9 |
| Kgatleng | Peri-Urban | 2.7 | 2.7 | 2.6 | 2.4 | 2.6 | 2.9 |
| North-East | Peri-Urban | 3.5 | 3.8 | 3.2 | 3.2 | 3.4 | 3.1 |
| Serowe Palapye | Semi-Urban | 3.3 | 3.5 | 3.1 | 3.0 | 3.2 | 3.1 |
| Central Mahalapye | Semi-Urban | 3.9 | 4.0 | 3.5 | 3.4 | 3.8 | 3.4 |
| Central Bobonong | Semi-Urban | 3.9 | 4.1 | 3.6 | 3.5 | 3.8 | 3.4 |
| Central Boteti | Semi-Urban | 3.4 | 3.7 | 3.3 | 3.2 | 3.2 | 3.6 |
| Central Tutume | Semi-Urban | 3.8 | 4.1 | 3.5 | 3.5 | 3.7 | 3.7 |
| Ngamiland East | Semi-Urban | 3.2 | 3.3 | 3.1 | 2.9 | 2.9 | 3.3 |
| Chobe | Semi-Urban | 2.5 | 2.4 | 2.5 | 2.3 | 2.2 | 3.2 |
| Barolong | Rural | 3.8 | 4.1 | 3.4 | 3.4 | 3.6 | 3.9 |
| Ngwaketse West | Rural | 3.9 | 4.0 | 3.5 | 3.5 | 3.7 | 3.6 |
| Kweneng West | Rural | 4.0 | 4.0 | 3.7 | 3.6 | 3.9 | 3.7 |
| Ngamiland West | Rural | 4.4 | 4.0 | 4.1 | 3.8 | 4.1 | 4.5 |
| Ghanzi | Rural | 3.1 | 3.1 | 3.0 | 2.8 | 2.8 | 3.2 |
| Kgalagadi South | Rural | 3.3 | 3.6 | 3.2 | 3.0 | 3.2 | 3.0 |
| Kgalagadi North | Rural | 2.9 | 3.0 | 2.9 | 2.7 | 2.7 | 2.8 |
| **Botswana** | — | 2.8 | 3.0 | 2.7 | 2.6 | 2.7 | 2.9 |

*Source: Authors' own calculations using results from Botswana's 2022 Population and Housing Census*

fertility levels occupy an intermediate demographic position, reflecting a more balanced distribution across young and older reproductive cohorts. Semi-urban districts such as Serowe–Palapye (TFR 3.1) and Central Mahalapye (3.4), along with peri-urban districts like Kweneng East (2.9), have significant concentrations of women in the 20–29 age group, yet also maintain sizable proportions in the 35–44 cohort.

### 3.5. Intersecting effects of mortality, women's socioeconomic status, and marital patterns in shaping regional fertility divergence in Botswana

The analysis of district-level fertility patterns in Botswana reveals strong associations between women's socioeconomic status, mortality outcomes, and TFR (See Table 3). Urban districts such as Gaborone, Francistown, Lobatse, Orapa, Jwaneng, and Sowa Town consistently exhibit low fertility levels (TFR 1.4–2.6), which align with broader indicators of social modernization. These areas are characterized by low infant and under-five mortality, high levels of female employment, and significant attainment of tertiary education. In Gaborone, for instance, nearly half of women have tertiary education (46.3%), and employment levels exceed 50%, contributing to the lowest fertility rates in the country (TFR 1.8). Sowa

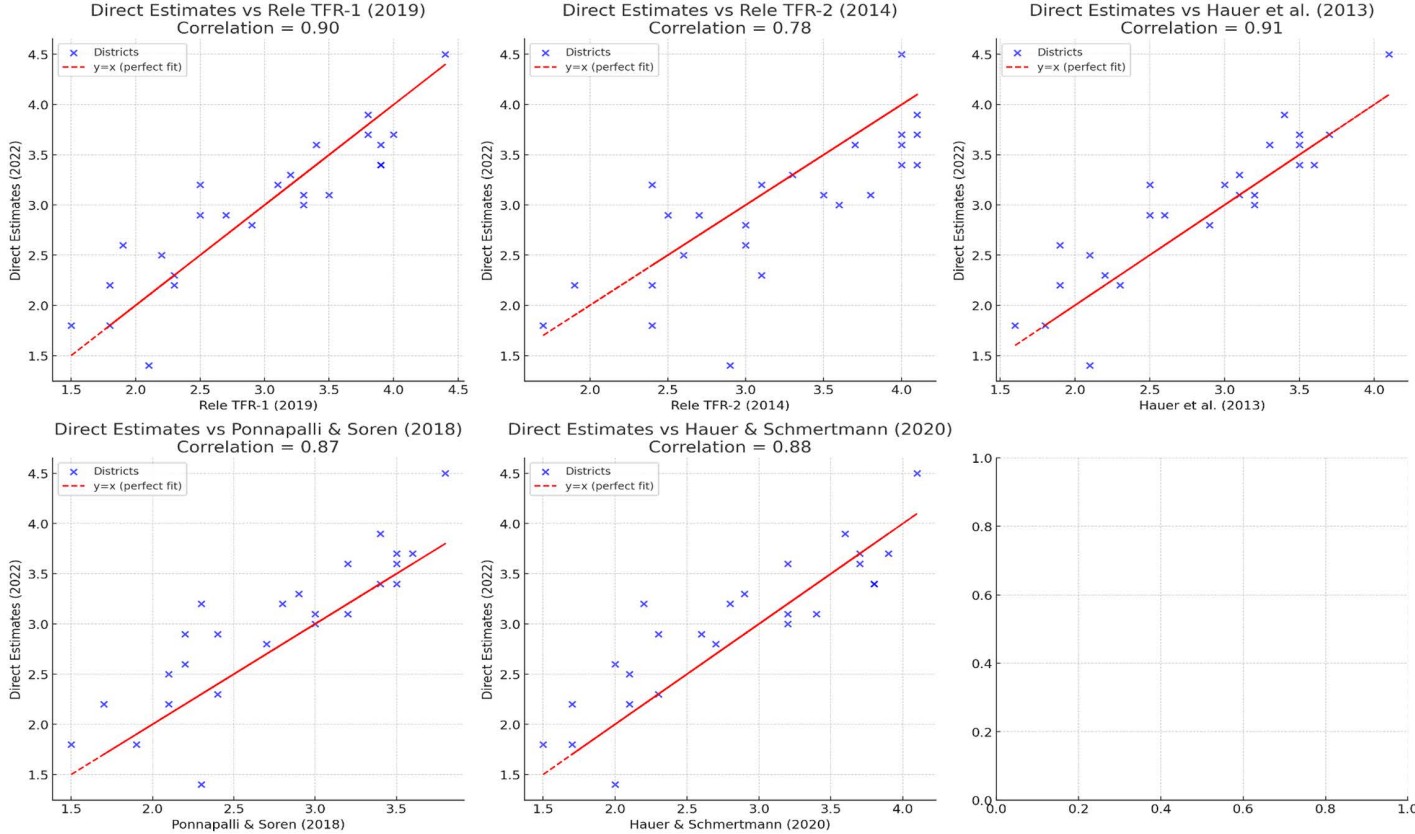

*Source: Authors' own calculations using results from Botswana's 2022 Population and Housing Census*

**Fig 3. Correlation between direct 2022 census-based fertility estimates and model-based estimates at the district level in Botswana, 2022.**

Town presents an even more striking case, with fertility suppressed to 1.4 despite high rates of marriage or cohabitation (39.5%).

By contrast, rural districts such as Barolong, Ngwaketse West, Central Mahalapye, Central Tutume and Ngamiland West are characterized by high mortality and limited female educational attainment, resulting in elevated fertility levels (TFR 3.4–4.5). For example, Ngamiland West records both the highest child mortality (U5MR 36) and the highest fertility (TFR 4.5), alongside one of the lowest levels of female tertiary education (12.7%). Here, mortality exerts a replacement effect, with families maintaining higher fertility as a strategy against the risk of child loss, while low education and restricted economic opportunities for women reinforce traditional reproductive norms. Similarly, Barolong exhibits high mortality and one of the lowest levels of tertiary education among women (15.4%), correlating with very high fertility (TFR 3.9).

Intermediate districts such as Kweneng East, Kgatleng, Serowe-Palapye, North-East, and Chobe exhibit moderate fertility levels (TFR 2.8–3.3), reflecting transitional dynamics. In these areas, female employment and education are improving but remain uneven, and mortality rates are higher than in urban centres. Chobe provides a noteworthy example: despite high mortality (U5MR 28), fertility is moderated to 3.2, largely due to female employment rates exceeding 50%. This case demonstrates that employment opportunities can act as a fertility-reducing factor, counterbalancing the upward pressure exerted by high mortality. Similarly, in Kweneng East and Kgatleng, moderate fertility (TFR 2.9) reflects an equilibrium between improving education levels and persisting traditional family formation practices.

**Table 2. Age distribution of women of reproductive age (15–49) and Its influence on district-level total fertility rates (TFR) in Botswana, 2022.**

| District | Settlement Type | 15–19 (%) | 20–24 (%) | 25–29 (%) | 30–34 (%) | 35–39 (%) | 40–44 (%) | 45–49 (%) | TFR |
|---|---|---|---|---|---|---|---|---|---|
| Gaborone | Urban | 13.38 | 17.97 | 16.39 | 15.41 | 14.97 | 12.64 | 9.47 | 1.8 |
| Francistown | Urban | 14.84 | 15.79 | 16.06 | 16.42 | 16.75 | 12.07 | 8.07 | 2.2 |
| Lobatse | Urban | 15.02 | 13.44 | 15.61 | 15.41 | 16.46 | 14.09 | 9.96 | 2.5 |
| Selebi-Pikwe | Urban | 17.35 | 11.52 | 14.11 | 14.55 | 16.47 | 14.01 | 10.44 | 2.3 |
| Orapa | Urban | 12.30 | 10.28 | 11.25 | 15.69 | 21.39 | 17.65 | 11.43 | 2.6 |
| Jwaneng | Urban | 14.97 | 15.08 | 19.56 | 20.51 | 22.51 | 17.52 | 11.46 | 1.8 |
| Sowa Town | Urban | 13.20 | 12.13 | 18.79 | 15.22 | 15.10 | 15.58 | 9.98 | 1.4 |
| South East | Peri-Urban | 13.13 | 19.32 | 17.10 | 15.37 | 14.00 | 11.76 | 9.35 | 2.2 |
| Kweneng East | Peri-Urban | 14.48 | 16.40 | 16.89 | 15.71 | 15.11 | 12.38 | 9.04 | 2.9 |
| Kgatleng | Peri-Urban | 15.27 | 15.17 | 15.86 | 15.18 | 15.13 | 13.17 | 10.23 | 2.9 |
| North East | Peri-Urban | 16.45 | 13.56 | 14.92 | 15.40 | 15.66 | 13.08 | 10.92 | 3.1 |
| Serowe–Palapye | Semi-Urban | 16.91 | 14.44 | 15.35 | 14.95 | 15.24 | 12.99 | 10.14 | 3.1 |
| Central Mahalapye | Semi-Urban | 17.31 | 13.27 | 14.09 | 14.74 | 15.61 | 13.79 | 11.18 | 3.4 |
| Central Bobonong | Semi-Urban | 17.03 | 13.22 | 14.89 | 14.13 | 15.44 | 13.62 | 11.66 | 3.4 |
| Central Boteti | Semi-Urban | 15.72 | 14.37 | 16.80 | 16.45 | 16.28 | 12.15 | 8.22 | 3.6 |
| Central Tutume | Semi-Urban | 17.50 | 14.11 | 15.03 | 14.96 | 15.07 | 12.44 | 10.90 | 3.7 |
| Ngamiland East | Semi-Urban | 16.15 | 15.36 | 17.38 | 17.88 | 16.08 | 11.86 | 8.44 | 3.3 |
| Chobe | Semi-Urban | 10.63 | 14.21 | 18.77 | 17.84 | 17.11 | 12.52 | 8.91 | 3.2 |
| Barolong | Rural | 17.05 | 13.69 | 14.74 | 14.35 | 15.32 | 13.10 | 11.76 | 3.9 |
| Ngwaketse West | Rural | 17.16 | 14.01 | 15.99 | 14.91 | 15.83 | 11.91 | 10.19 | 3.6 |
| Kweneng West | Rural | 16.99 | 15.04 | 15.46 | 15.02 | 15.72 | 11.95 | 9.81 | 3.7 |
| Ngamiland West | Rural | 16.60 | 15.12 | 17.49 | 15.56 | 15.30 | 11.53 | 8.43 | 4.5 |
| Ghanzi | Rural | 14.98 | 16.18 | 16.89 | 17.87 | 14.67 | 10.54 | 8.87 | 3.2 |
| Kgalagadi South | Rural | 17.18 | 13.53 | 15.70 | 14.65 | 14.89 | 12.63 | 11.42 | 3.0 |
| Kgalagadi North | Rural | 17.51 | 16.32 | 22.17 | 18.45 | 18.89 | 15.29 | 11.20 | 2.8 |

*Source: Authors' own calculations using results from Botswana's 2022 Population and Housing Census*

## 4. Discussion and conclusion

This study set out to estimate district-level fertility from age-structured census data and assess spatial–socioeconomic differentials in Botswana using the 2022 Population and Housing Census. Three core objectives guided the analysis: (i) to generate transparent district-level Total Fertility Rate (TFR) estimates using direct census methods, (ii) to validate these estimates through comparisons with multiple indirect model-based approaches, and (iii) to examine how mortality, women's socioeconomic status, and marital patterns interact to shape fertility variation across districts.

### 4.1. Critical discussion of findings

A key finding of the study is the strong alignment between direct census-based fertility estimates and indirect model-based measures at the national level, with TFR values converging between 2.6 and 3.0 children per woman. This convergence provides compelling evidence that Botswana has entered an advanced stage of fertility transition and, critically, that the 2022 census constitutes a reliable basis for fertility estimation at both national and subnational scales. The consistency observed across estimation approaches reinforces confidence in the robustness of census-derived fertility indicators and supports their use for routine demographic monitoring.

At the same time, the analysis reveals substantial and demographically meaningful district-level variation, which is obscured by national averages. Urban centres such as Gaborone, Francistown, and mining towns including Jwaneng

**Table 3. District-level associations between mortality, women's socioeconomic characteristics, and total fertility rates (TFR) in Botswana, 2022.**

| District | Settlement Type | IMR | U5MR | % Women Employed | % Women Married/Cohabiting | % Women Tertiary Education | TFR | Interpretation |
|---|---|---|---|---|---|---|---|---|
| Gaborone City | Urban | 8 | 9 | 50.7 | 30.1 | 46.3 | 1.8 | Low mortality, high employment and education, low union prevalence→lowest TFR |
| Francistown City | Urban | 12 | 14 | 48.3 | 31.4 | 28.7 | 2.2 | Moderate mortality, high employment, moderate education→low TFR |
| Lobatse | Urban | 10 | 11 | 48.9 | 32.0 | 29.0 | 2.5 | Low mortality, high employment→low fertility |
| Selebi Phikwe | Urban | 21 | 24 | 46.8 | 37.2 | 27.1 | 2.3 | Higher mortality slightly increases TFR, despite moderate employment and education |
| Orapa | Urban | 0 | 4 | 58.9 | 40.7 | 43.4 | 2.6 | Very low mortality, high employment and education→moderate TFR due to migration and family formation dynamics |
| Jwaneng | Urban | 18 | 18 | 56.2 | 37.9 | 39.0 | 1.8 | moderate mortality, high employment/education→low TFR |
| Sowa Town | Urban | 0 | 0 | 57.8 | 39.5 | 39.9 | 1.4 | Extremely low mortality, high employment/education→lowest TFR nationally |
| South-East | Peri-Urban | 8 | 11 | 46.6 | 25.6 | 41.9 | 2.2 | Low mortality, high employment and education→low TFR |
| Kweneng East | Peri-Urban | 11 | 14 | 36.9 | 24.1 | 26.3 | 2.9 | Moderate mortality, moderate education/employment→mid-range TFR |
| North-East | Peri-Urban | 17 | 18 | 44.0 | 34.6 | 22.3 | 3.1 | Moderate mortality, moderate education/employment→slightly above average TFR |
| Kgatleng | Peri-Urban | 12 | 14 | 39.4 | 26.0 | 26.8 | 2.9 | Moderate mortality and education→moderate TFR |
| Serowe Palapye | Semi-Urban | 15 | 19 | 34.8 | 20.7 | 20.8 | 3.1 | Moderate mortality, low employment/education→slightly high TFR |
| Central Mahalapye | Semi-Urban | 23 | 28 | 33.2 | 20.8 | 15.6 | 3.4 | High mortality, low education→high TFR |
| Central Bobonong | Semi-Urban | 28 | 34 | 41.7 | 28.2 | 16.8 | 3.4 | Very high mortality, moderate employment→high TFR |
| Central Boteti | Semi-Urban | 14 | 19 | 36.2 | 26.5 | 18.3 | 3.6 | Moderate mortality, low education→high TFR |
| Central Tutume | Semi-Urban | 21 | 25 | 35.9 | 24.4 | 16.1 | 3.7 | High mortality, low education/employment→high TFR |
| Ngamiland East | Semi-Urban | 18 | 23 | 37.9 | 28.0 | 23.3 | 3.3 | Moderate mortality, low-medium education→higher TFR |
| Chobe | Semi-Urban | 23 | 28 | 54.4 | 35.9 | 23.2 | 3.2 | High employment moderates fertility despite high mortality→TFR slightly above national average |
| Ngamiland West | Rural | 29 | 36 | 34.8 | 31.6 | 12.7 | 4.5 | Highest mortality, low employment/education→highest TFR nationally |
| Kweneng West | Rural | 23 | 26 | 32.9 | 31.7 | 11.3 | 3.7 | High mortality, low education→high TFR |
| Barolong | Rural | 21 | 25 | 32.4 | 30.5 | 15.4 | 3.9 | High mortality, low employment and education→very high TFR |
| Ngwaketse West | Rural | 24 | 28 | 33.3 | 21.8 | 12.6 | 3.6 | High mortality, low education/employment→high TFR |
| Ghanzi | Rural | 25 | 33 | 40.8 | 36.2 | 20.1 | 3.2 | High mortality, moderate education/employment→high TFR |
| Kgalagadi South | Rural | 22 | 25 | 38.3 | 23.1 | 15.6 | 3.0 | High mortality, low education → above-average TFR |
| Kgalagadi North | Rural | 25 | 28 | 48.2 | 28.2 | 21.4 | 2.8 | High employment reduces fertility moderately→TFR slightly below rural averages |

*Source: Authors' own calculations using results from Botswana's 2022 Population and Housing Census*

and Sowa Town record fertility levels well below replacement, reflecting delayed childbearing, high female educational attainment, and strong labour market participation. In contrast, rural districts such as Ngamiland West, Barolong, and Central Mahalapye exhibit persistently high fertility, with TFRs ranging from approximately 3.5 to 4.5. These differences are significant not only in magnitude but also in implication, as they signal divergent demographic trajectories within the same national context.

The persistence of high fertility in rural districts is closely associated with elevated child mortality, limited women's socioeconomic empowerment, and enduring traditional reproductive norms. Conversely, districts characterised by lower fertility tend to exhibit older reproductive-age structures, lower mortality, higher levels of tertiary education, and greater female employment. Transitional districts display intermediate fertility profiles, suggesting that socioeconomic change moderates fertility behaviour without immediately erasing the influence of mortality and early childbearing. These findings extend earlier national-level evidence of fertility decline in Botswana documented by Bainame and Letamo [5] by demonstrating that the fertility transition is spatially uneven and socially stratified.

The observed spatial patterns are consistent with broader evidence from sub-Saharan Africa and other developing contexts, where fertility transitions progress unevenly across rural–urban and socioeconomic divides [18,19]. In particular, the association between low fertility and women's education and employment mirrors findings from South Africa [20], while the persistence of high fertility in districts with elevated child mortality aligns with evidence from East African settings, where replacement behaviour sustains higher fertility levels [21].

From a methodological perspective, an important contribution of this study lies in its explicit triangulation of direct and indirect fertility estimates. While convergence across methods at the national level strengthens confidence in the overall fertility regime, divergences at the district level, especially in small-population districts such as Sowa Town, highlight the limitations of indirect models when applied to finely disaggregated geographic units. In such contexts, model-based approaches may smooth or inflate fertility estimates due to assumptions of population stability and regular age structures. These divergences are therefore analytically informative, underscoring that indirect models are most effective as validation tools rather than substitutes for direct census estimation when district-level precision is required

## 4.2. Policy implications

The findings carry significant policy relevance. First, the persistent urban–rural fertility divide calls for targeted interventions that acknowledge regional heterogeneity. In rural districts where fertility remains high, policies should prioritize investments in female education, economic empowerment, and reproductive health services, including improved access to family planning. These measures are likely to accelerate the transition towards smaller family sizes.

Second, the association between child mortality and high fertility highlights the urgency of further reducing under-five mortality through investments in child health, nutrition, and healthcare access. Mortality reduction can break the cycle of replacement behaviour that sustains high fertility in disadvantaged districts.

Third, in urban centres where fertility is below replacement, policymakers must anticipate potential long-term implications, including population ageing, labour force contraction, and rising dependency ratios. This calls for forward-looking strategies such as supporting work–family balance, encouraging sustainable family sizes, and adapting social and economic policies to mitigate demographic shifts.

Finally, the demonstrated utility of census-based fertility estimation underscores the need to institutionalize regular district-level demographic monitoring, integrating direct census estimates with innovative model-based approaches. This would strengthen evidence-based planning in education, health, housing, and labour market development, ensuring interventions are aligned with the demographic realities of each district.

### 4.3. Limitations

Despite its contributions, the study is not without limitations. First, census-based fertility data remain vulnerable to recall bias, proxy reporting, and reference-period errors, which may understate or misclassify births. Second, small population districts introduce volatility into direct estimates, as observed in Sowa Town. Third, the cross-sectional nature of census data limits causal inference; while associations between fertility and socioeconomic factors are observed, longitudinal data would be needed to confirm directionality and dynamics over time. Finally, indirect methods rely on assumptions of stable populations and uniform mortality, which may not hold in contexts of rapid demographic transition or internal migration.

### 4.4. Conclusion

This study provides one of the most comprehensive assessments of district-level fertility in Botswana to date. By combining direct census estimation with comparative model-based techniques, it demonstrates the reliability of census data while capturing the heterogeneity of fertility across the country. The results reveal a Botswana firmly in transition: urban districts approaching or falling below replacement fertility, rural districts sustaining high fertility linked to mortality and socioeconomic disadvantage, and transitional areas reflecting mixed trajectories.

The implications are clear: fertility decline in Botswana is uneven, shaped by structural inequalities in education, employment, and health. Policies must therefore adopt a differentiated approach, targeting rural areas with interventions to reduce mortality and expand women's opportunities, while preparing urban areas for the socioeconomic consequences of very low fertility. Future research should integrate longitudinal and qualitative perspectives to better understand the behavioural drivers of fertility decisions at district level. Ultimately, the study highlights that achieving equitable fertility transitions across Botswana will require not only demographic monitoring but also sustained social investment that addresses the root determinants of reproductive behaviour.

## Author contributions

**Conceptualization:** Tiro Theodore Monamo.

**Formal analysis:** Tiro Theodore Monamo.

**Investigation:** Tiro Theodore Monamo.

**Resources:** Kannan Navaneetham.

**Supervision:** Kannan Navaneetham.

**Validation:** Kannan Navaneetham.

**Writing – original draft:** Tiro Theodore Monamo.

**Writing – review & editing:** Tiro Theodore Monamo.

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
