## [Decision Letter · Decision Letter 0]

2 Dec 2025

Dear Dr. Monamo,

Thank you for submitting your manuscript to PLOS ONE. After careful consideration, we feel that it has merit but does not fully meet PLOS ONE’s publication criteria as it currently stands. Therefore, we invite you to submit a revised version of the manuscript that addresses the points raised during the review process.

We look forward to receiving your revised manuscript.

Kind regards,

Alfredo Luis Fort, M.D., M.Sc., Ph.D.

Academic Editor

PLOS ONE

Journal Requirements:

4. In the online submission form, you indicated that the data supporting the findings of this study are accessible through Statistics Botswana and can be obtained by researchers upon request. Additionally, the data may be provided by the authors upon reasonable request, subject to approval from Statistics Botswana.

5. Please ensure that you refer to Figure 2 in your text as, if accepted, production will need this reference to link the reader to the figure.

6. Please upload a copy of Figure 1, and 3, to which you refer in your text on page 5, 6, and 14. If the figure is no longer to be included as part of the submission please remove all reference to it within the text.

7. Please include a copy of Table 1 which you refer to in your text on page 10, 12, and 19.

8. We note you have included a table to which you do not refer in the text of your manuscript. Please ensure that you refer to Table 2 in your text; if accepted, production will need this reference to link the reader to the Table.

Additional Editor Comments :

This is a well designed study and a manuscript that fulfills most requirements for publication. However, there are areas that needs more or better description, including improving graphs and tables. You will find a number of suggestions in the letter as well as attached file for the improvement of the manuscript in order to get it ready for publication. Thanks.

Reviewers' comments:

Reviewer's Responses to Questions

**Comments to the Author**

1. Is the manuscript technically sound, and do the data support the conclusions?

Reviewer #1: Yes

2. Has the statistical analysis been performed appropriately and rigorously?

Reviewer #1: No

3. Have the authors made all data underlying the findings in their manuscript fully available?

Reviewer #1: Yes

4. Is the manuscript presented in an intelligible fashion and written in standard English?

Reviewer #1: Yes

Reviewer #1: Abstract

The abstract is missing a clear presentation of standard demographic procedures, key findings, and comparisons between direct estimates and model-based fertility measures and summarized conclusions. The abstract need to reflect the analytical scope of the paper.

Introduction

The demographic context in Botswana should, which is explicitly highlighted in the methods, may be could be more communicative in the introduction part.

The paragraph beginning with “The 2022 PHC introduced major methodological innovations…” should also be moved to the Introduction, not the Methods section, because it provides contextual background rather than describing analytic procedures.

The paragraph beginning with “The contribution of this study is both methodological and substantive…” is appropriate as the final paragraph of the Introduction.

A reference to WHO should be added where socioeconomic determinants are defined or classified.

Methodology

Information that presents results, outputs, or interpretation should not appear in the Methods section. Any narrative explaining trends, district differences, or preliminary data outcomes should be moved to Findings or Discussion.

Mitigation measures to quality issues highlighted in the paper should come out more specific and explicit.

Findings

Only empirical results should appear in the Findings section. Several passages drift into interpretation and should be moved:

Comparisons of direct outputs with model-based methods, for instance a paragraph which starts “Similar patterns are evident in Francistown…” suits well in the discussion, not findings.

Statements linking socioeconomic factors and fertility, for instance “The analysis reveals strong associations between women’s socioeconomic status, mortality outcomes, and fertility…” requires actual statistical justification e.g., correlations, regressions, effect sizes.

Some findings currently read as conclusions, for example, for example “These findings reinforce the utility of combining methods…”. This would suit the conclusion.

Figure 1 contains an empty or non-interpretable output and should be removed, as it may confuse readers.

Table 2 does not clearly demonstrate the proposed relationship between age distribution and TFR. The table needs either a statistical test showing influence, or restructuring so the relationship becomes interpretable.

Other comments

The manuscript repeatedly restates the objectives in several sections. This creates redundancy and should be streamlined.

The summary and conclusion should be reinforced with clearer statements specifically on significance of variations and implications of direct vs. model-based differences,

Summary:

The paper demonstrates strong methodological potential and uses rigorous demographic techniques. However, sections are not well differentiated, resulting in conceptual mixing of methods, findings, discussion, and conclusions. Strengthening the analytical presentation, improving statistical justification, and repositioning content within appropriate sections will significantly enhance the paper’s coherence and scientific credibility.

**Do you want your identity to be public for this peer review?** For information about this choice, including consent withdrawal, please see our Privacy Policy

Reviewer #1: **Yes:** Abel Mokua Nyabera

---

## [Author Response · Author response to Decision Letter 1]

13 Jan 2026

Estimation of District-Level Fertility using Age-Structured Census Data and Assessment of Spatial–Socioeconomic Differentials in Botswana, 2022

Response to Reviewer #1

We sincerely thank Reviewer #1 for the careful and constructive evaluation of our manuscript. The comments were instrumental in improving the clarity, structure, and scientific rigour of the paper. Below, we provide a detailed response to each comment and describe the specific revisions made in the amended manuscript.

REVIEWER’S COMMENTS RESEARCHER’S RESPONSE

1. ABSRACT

The abstract is missing a clear presentation of standard demographic procedures, key findings, and comparisons between direct estimates and model-based fertility measures and summarized conclusions. The abstract need to reflect the analytical scope of the paper. We fully agree with this assessment.

The abstract has been substantially revised to clearly reflect the analytical scope of the paper. Specifically:

• Standard demographic procedures are now explicitly stated, including the calculation of ASFRs and TFRs from census data.

• Comparative validation against established indirect and model-based methods (Rele; Hauer et al.; Ponnapalli & Soren; Hauer & Schmertmann) is clearly articulated.

• Key empirical findings are summarised, including national-level convergence (TFR 2.6–3.0) and pronounced district-level heterogeneity.

• Conclusions and policy implications are now explicitly stated.

These revisions ensure that the abstract now provides a complete, self-contained summary of methods, findings, and implications.

2. Introduction

The demographic context in Botswana should, which is explicitly highlighted in the methods, may be could be more communicative in the introduction part.

We have strengthened the demographic context in the Introduction by:

• Expanding discussion of Botswana’s long-term fertility transition using historical evidence (Bainame & Letamo).

• Explicitly situating district-level heterogeneity within broader processes of urbanisation, education expansion, and labour force participation.

• Clarifying why national averages obscure meaningful subnational fertility dynamics.

This ensures that the demographic motivation for the study is clearly established before the Methods section.

The paragraph beginning with “The 2022 PHC introduced major methodological innovations…” should also be moved to the Introduction.

This paragraph has been relocated from the Methods to the Introduction, where it now provides contextual background on data quality, digital enumeration, and census reliability. The Methods section is now strictly confined to analytical procedures.

The paragraph beginning with “The contribution of this study is both methodological and substantive…” is appropriate as the final paragraph of the Introduction. We agree and have retained this paragraph as the final paragraph of the Introduction, unchanged in substance but now better positioned following the expanded contextual discussion.

A reference to WHO should be added where socioeconomic determinants are defined or classified. A reference to WHO frameworks has now been incorporated in the Introduction where socioeconomic determinants (education, employment, marital patterns, and access to services) are conceptually defined. This strengthens the theoretical grounding and aligns the study with established public health classifications.

3. Methodology

Information that presents results, outputs, or interpretation should not appear in the Methods section.

We have carefully reviewed the Methods section and removed all interpretive or results-oriented statements. The section now strictly describes:

• Data sources

• Quality assessment procedures

• Estimation techniques

• Validation strategy

• Measurement of socioeconomic and mortality indicators

All interpretation and trend discussion has been relocated to the Findings or Discussion sections.

Mitigation measures to quality issues highlighted in the paper should come out more specific and explicit.

This issue has been addressed by expanding Section 2.2 (Data Quality Assessment and Mitigation Measures). The revised section now explicitly details:

• Parity and recent-birth consistency checks

• Aggregation strategies for small districts

• Triangulation across multiple indirect methods

• Criteria used to prioritise estimates where discrepancies arise

• Cautious interpretation in migration-affected districts

These additions clarify how potential biases were actively managed rather than merely acknowledged.

4. Findings

Only empirical results should appear in the Findings section.

We have carefully revised the Findings section to ensure that it now reports only empirical results. Specifically:

• Interpretive comparisons (e.g., “Similar patterns are evident in Francistown…”) have been moved to the Discussion.

• Normative or evaluative statements have been removed from the Findings.

Statements linking socioeconomic factors and fertility require statistical justification.

We agree and have strengthened this section by:

• Explicitly grounding statements in observed district-level associations presented in Table 3.

• Clarifying that relationships are descriptive and correlational, consistent with census-based cross-sectional data.

• Reserving causal interpretation for the Discussion, where findings are situated within demographic transition theory and comparative literature.

Some findings currently read as conclusions.

Statements such as “These findings reinforce the utility of combining methods…” have been relocated to the Discussion and Conclusion, ensuring that the Findings section remains strictly empirical.

Table 2 does not clearly demonstrate the proposed relationship between age distribution and TFR.

Table 2 has been restructured, and its interpretation clarified in the text to explicitly link age distribution patterns with observed TFR variation across districts. The discussion now focuses on percentage distributions and demographic composition rather than implicit inference.

The manuscript repeatedly restates the objectives.

We have streamlined the manuscript by:

• Stating the objectives once clearly in the Introduction

• Referring back to them implicitly rather than restating them verbatim in later sections

This has reduced redundancy and improved narrative flow.

6. Summary and Conclusion

The summary and conclusion should be reinforced with clearer statements on variation significance and direct vs. model-based differences.

The Discussion and Conclusion have been strengthened to:

• Clearly articulate the significance of district-level variation

• Explicitly discuss where and why direct and model-based estimates diverge

• Highlight implications for policy, planning, and demographic monitoring

7. Overall Assessment

Strengthening analytical presentation and repositioning content will enhance coherence and credibility.

We appreciate this summary and believe the revisions directly address these concerns. The manuscript now demonstrates:

• Clear separation of Methods, Findings, and Discussion

• Stronger statistical and demographic justification

• Improved coherence and scientific credibility

We are grateful for the reviewer’s insightful guidance, which has substantially improved the manuscript.

---

## [Editor Report · Decision Letter 1]

19 Jan 2026

Estimation of District-Level Fertility using Age-Structured Census Data and Assessment of Spatial–Socioeconomic Differentials in Botswana, 2022

PONE-D-25-55917R1

Dear Dr. Monamo,

We’re pleased to inform you that your manuscript has been judged scientifically suitable for publication and will be formally accepted for publication once it meets all outstanding technical requirements.

Kind regards,

Alfredo Luis Fort, M.D., M.Sc., Ph.D.

Academic Editor

PLOS One

Additional Editor Comments:

The authors have made the necessary changes and additions suggested by the reviewer and the editor, which makes the manuscript publishable. However, there are some very minor questions/suggestions (in the attached file) that the main editor can suggest the authors to undertake while preparing for publication. Thanks.

---

## [Editor Report · Acceptance letter]

PONE-D-25-55917R1

PLOS One

Dear Dr. Monamo,

I'm pleased to inform you that your manuscript has been deemed suitable for publication in PLOS One. Congratulations! Your manuscript is now being handed over to our production team.

Kind regards,

on behalf of

Dr. Alfredo Luis Fort

Academic Editor

PLOS One